# Scalable Continuous-time Diffusion Framework for Network Inference and Influence Estimation

## ABSTRACT

The study of continuous-time information diffusion has been an important area of research for many Web applications in recent years. When only the diffusion traces (cascades) are accessible, cascade-based network inference and influence estimation are two essential problems to explore. Alas, existing methods exhibit limited capability to infer and process networks with more than a few thousand nodes, suffering from scalability issues. In this paper, we view the diffusion process as a *continuous-time dynamical system*, based on which we establish a continuous-time diffusion model. Subsequently, we instantiate the model to a scalable and effective framework (FIM) to approximate the diffusion propagation from available cascades, thereby inferring the underlying network structure. Furthermore, we undertake an analysis of the approximation error of FIM for network inference. To achieve the desired scalability for influence estimation, we devise an advanced sampling technique and significantly boost the efficiency. We also quantify the effect of the approximation error on influence estimation theoretically. Experimental results showcase the effectiveness and superior scalability of FIM on network inference and influence estimation.

## KEYWORDS

Network Inference, Influence Estimation, Continuous-time Dynamical System

**ACM Reference Format:**
Anonymous Author(s). 2024. Scalable Continuous-time Diffusion Framework for Network Inference and Influence Estimation. In *Proceedings of Proceedings of the ACM Web Conference 2023 (WWW '24)*. ACM, New York, NY, USA, 12 pages.

## 1 INTRODUCTION

Over the past few decades, there has been extensive interest in the study of continuous-time information diffusion [3, 13, 32, 40]. Information of various kinds (tweets, memes, likes, etc) spreads along the established connections in heterogeneous networks on the Web, by following specific diffusion models. For example, a piece of information may be posted / reposted, with a timestamp, by users with followee-follower relationships in an online social network (OSN) such as Twitter or Weibo. That piece of information may spread *virally*, and its diffusion cascade is traceable by inspecting the posting timestamps of the associated users. In particular, the user who starts the post is the *seed user* and the others are the

activated users (reposters). Continuous-time diffusion models are often used to characterize such cascades initiated by seed users in social networks, assuming knowledge of the underlying diffusion network and of the associated influence probability for each connection. However, directly obtaining and exploiting this knowledge about the underlying diffusion medium – as some state-of-the-art methods assume [12, 31] – is often impossible in reality. Instead, an abundance of continuous-time cascade data can be collected, recording the source users and the activated ones, with corresponding timestamps. While the activation times are usually recorded, observing the individual transmissions and the transmission paths they induce is typically harder. Therefore, a prevalent challenge is to infer the diffusion network from continuous-time cascade data that only provides such activation times. Additionally, estimating the influence or spread potential of a given seed user or set thereof in the inferred network is equally challenging and important, in order to answer influence maximization (IM) queries.

In the literature, a plethora of methods have been developed for network inference and influence estimation based solely on continuous-time cascades. Specifically, NETINF [12] and ConNIe [30] leverage maximum likelihood estimation (MLE) for cascade modeling and optimize the diffusion model via submodular maximization and convex optimization respectively. Different from NETINF and ConNIe, which assume the transmission rate for all edges to be the same, NetRate [11] allows for temporally heterogeneous diffusions and proposes an elegant solution based on convex programming. Finally, the recent work of [18] proposed a neural mean-field framework (NMF) for network inference and influence estimation. By assuming access to ground-truth networks, InfluMax [14] and ConTinEst [9] devise advanced sampling techniques for influence estimation. (More discussions on the related research are deferred to Section 6). However, current methods are evaluated on (and can only cope with) networks at small scale, with at most order-of-thousands nodes, and face significant scalability issues, as shown in our experiments.

In order to address these limitations of the state-of-the-art, we aim to build a simple yet effective framework for network inference and influence estimation based on cascades, able to scale to real-world networks. To begin with, observe that online users usually get more exposed to marketing information as this is initially diffused in their community, and the influence of neighbors on that specific piece of information typically diminishes gradually with time. This is the well-known *market saturation* effect [24, 25, 36, 38, 42]. To model this phenomenon, we employ exponential functions with learnable parameters to quantify the transmission rate between users [18, 39]. In epidemiological models, the utilization of exponential distributions to model waiting times is a natural assumption [1, 2, 29, 34]. Furthermore, we view the influence diffusion process as a *continuous-time dynamical system* (CDS) [4, 22] and each user as a *particle* therein. We leverage the CDS abstraction to model the continuous-time diffusion propagation. Upon the established

model, we propose the **F**ramework of d**I**ffusion approxi**M**ation (FIM) to estimate diffusion parameters by reconstructing the propagations from cascade data, thereby inferring the underlying network structures. Subsequently, to achieve the desired efficiency for influence estimation, we propose the sampling technique *shortest diffusion time of set* (SDTS). We analyze the network inference error of FIM and quantify the effect of such errors on the influence estimation theoretically. Our experiments on synthetic and real-world datasets demonstrate that (i) the FIM approach significantly improves the state-of-the-art learning ability on network inference, and (ii) it significantly boosts the efficiency of influence estimation, scaling to realistic datasets. In particular, our method is the only one able to tackle real-world networks with $12,677$ nodes with thousands of cascades, while offering superior performance.

In a nutshell, our contributions can be summarized as follows.

- We develop a model for continuous-time diffusion and design a scalable and effective framework (FIM) to address network inference and influence estimation.
- We systematically analyze the approximation errors of FIM for both network inference and influence estimation.
- We enhance the scalability of influence estimation, by a sampling technique to estimate the shortest diffusion time of sets.
- We run experiments on synthetic and real-world data, which confirms the superior scalability and effectiveness of FIM.

## 2 PRELIMINARY

**Table 1: Frequently used notations**

| Notation | Description |
|---|---|
| $G = (\mathcal{V}, \mathcal{E})$ | a network with node set $\mathcal{V}$ and edge set $\mathcal{E}$ |
| $n, m$ | the number of nodes and the number edges in $G$ |
| $\mathcal{N}_u, d_u$ | the incoming neighbor set of node $u$ and its in-degree |
| $A$ | the adjacency parameter matrix of $G$ |
| $c, C$ | a cascade recording the activation times for nodes in $\mathcal{V}$ and the set of all known cascades |
| $h_t, \mathcal{H}_t$ | the specific observation of cascade $c$ and the observation of a random cascade from $C$ at time $t$ |
| $\phi_t, \Phi_t$ | the specific node state in observation $h_t$ and the random node state in random observation $\mathcal{H}_t$ |
| $\gamma_t, \Gamma_t$ | the diffusion rates of nodes in cascade $c$ and the expected diffusion rates of nodes at time $t$ |
| $\odot$ | the Hadamard product |

### 2.1 Diffusion Networks

**Notations.** We use calligraphic fonts, bold uppercase letters, and bold lowercase letters to represent sets (e.g., $\mathcal{N}$), matrices (e.g., $A$), and vectors (e.g., $c$) respectively. The $i$-th row (resp. column) of matrix $A$ is denoted by $A[i, \cdot]$ (resp. $A[\cdot, i]$). For ease of exposition, node $u$ can indicate the row (resp. column) associated with $u$ in the matrix, e.g., $A[u, \cdot]$ (resp. $A[\cdot, u]$). Frequently used notations are summarized in Table 1.

Let $G = (\mathcal{V}, \mathcal{E})$ be a directed graph with nodes $\mathcal{V}$ and edges $\mathcal{E}$, and let $n = |\mathcal{V}|$ and $m = |\mathcal{E}|$. We assume that the diffusion process along edge $\langle u, v \rangle \in \mathcal{E}$ from $u$ to $v$ is determined by a diffusion probability density function (PDF) $p(\lambda_{uv}, t) = \begin{cases} \lambda_{uv} e^{-\lambda_{uv} t} & \text{if } t \geq 0 \\ 0 & \text{if } t < 0 \end{cases}$

where $\lambda_{uv} \in [0, \infty)$ is the *diffusion rate* determining the strength of node $u$ influencing $v$ and $t$ is the diffusion delay. For example, Figure 1 illustrates the expanding tendency of a diffusion PDF with $\lambda_{uv} = 2$ with time. Accordingly, the cumulative distribution function (CDF) $F(\lambda_{uv}, t) = \int_0^t p(\lambda_{uv}, t')dt' = 1 - e^{-\lambda_{uv} t}$ $(t \geq 0)$ is the probability that node $u$ succeeds in influencing $v$ along edge $\langle u, v \rangle$ by time $t$. Let $A$ be the *adjacency parameter* matrix of $G$ with $A[u, v] = \lambda_{uv}$ if $\langle u, v \rangle \in \mathcal{E}$ and $A[u, v] = 0$ otherwise, and let $\mathcal{N}_u$ be the direct *incoming* neighbor set of $u$ with *in-degree* $d_u = |\mathcal{N}_u|$.

### 2.2 Continuous-time Diffusion Model

The independent cascade (IC) and linear threshold (LT) models [23] are well-adopted discrete-time models for vanilla IM. Yet discrete-time models have a synchronization property by nature and have been shown empirically to have limitations for modeling diffusion processes in reality [10, 14, 15, 21, 35]. Therefore, we adopt here the *continuous-time independent cascade* (CIC) model for diffusion between users [8, 9, 11, 16, 18]. Given a time window $T > 0$ and a source set $\mathcal{S} \subset \mathcal{V}$, the influence from $\mathcal{S}$ under the CIC model stochastically spreads as follows.

At time $t = 0$, all nodes $u \in \mathcal{S}$ are activated. When a node $u$ is activated at time $t_u$, it attempts to influence each of its outgoing and inactive neighbors $v$ by following the diffusion function $p(\lambda_{uv}, t - t_u)$. Each node $v$ is activated by its incoming neighbors *independently*. Once activated, $v$ remains active and tries to influence its outgoing neighbors. A diffusion process reaches a fixed point ultimately and ends at time $T$. In what follows, To avoid clutter in notations, we do not specify the source set $\mathcal{S}$ nor the time window $T$ (if applicable) when these are clear from the context.

A diffusion process originating at the source (seed) set $\mathcal{S}$ and ending by time $T$, which underwent as just described, is denoted as a *cascade* $c := \{t_u : \forall u \in \mathcal{V}\} \in \mathbb{R}_{\geq 0}^n$ where $t_u \in [0, T] \cup \{\infty\}$ is the activation time of node $u$. In particular, $t_u := \infty$ if node $u$ is not activated within the time window $T$. In applications, for previously recorded cascades, the source set $\mathcal{S}$ may not be given explicitly but can be easily inferred from any cascade $c$ as $\mathcal{S} := \{u : \forall t_u \in c, t_u = 0\}$; $\mathcal{S}$ may be a singleton. For diffusions initialized by $\mathcal{S}$, let $C$ be the set of all the known cascades from $\mathcal{S}$ within the time window $T$. Given a cascade $c$ and time stamp $t \in [0, T]$, let $h_t \in \mathbb{R}_{\geq 0}^n$ denote the *observation* of $c$ at time $t$, i.e., $h_{t,u} := t_u$ for $t_u \in c$ if $t_u \leq t$; otherwise $h_{t,u} := \infty$. Therefore, $h_t$ is the snapshot of cascade $c$ at time $t$. In general, let the variable $\mathcal{H}_t$ be the observation (random vector) of cascades randomly sampled from $C$. To quantify the probability of each node being activated by time $t$, the states of nodes during diffusion at time $t$ are recorded. Specifically, given an observation $h_t$, let $\phi_t \in \{0, 1\}^n$ be the corresponding *state* vector of nodes in $\mathcal{V}$, i.e., $\phi_{t,v} = 1$ if $h_{t,v} \leq t$ and $\phi_{t,v} = 0$ otherwise. Accordingly, let $\Phi_t$ be the corresponding random state vector of $\mathcal{H}_t$. Therefore, $\mathbb{E}[\Phi_t] = \mathbb{E}_{\mathcal{H}_t \sim C}[\Phi_t]$ indicates the probability vector of nodes in $\mathcal{V}$ activated by time $t$.

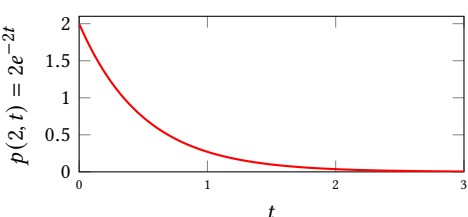

**Figure 1: Diffusion function $p(2, t)$.**

For example, Figure 2 illustrates a cascade within time $T$, which is $\mathbf{c} = \{t_0, t_1, \cdots, t_7, t_8\}$, with $0 = t_0 < t_1 < t_3 < t_4 < t_5 < T$ and $t_{\{2,6,7,8\}} = \infty$. By taking an observation at some time moment $t^* \in (t_4, t_5)$, we have $0 = \mathbf{h}_{t^*,0} < \mathbf{h}_{t^*,1} < \mathbf{h}_{t^*,3} < \mathbf{h}_{t^*,4} < t^*$ and $\mathbf{h}_{t^*,\{2,5,6,7,8\}} = \infty$. Correspondingly, we have $\phi_{t^*,\{0,1,3,4\}} = 1$ and $\phi_{t^*,\{2,5,6,7,8\}} = 0$.

Particularly, the formal definitions for the problems of **network inference** and **influence estimation** are as follows.

**DEFINITION 1 (NETWORK INFERENCE).** *Let* $\mathbf{G} = (\mathcal{V}, \mathcal{E})$ *be a diffusion network with a known node set* $\mathcal{V}$, $|\mathcal{V}| = n$, *an unknown edge set* $\mathcal{E}$, *and an unknown associated adjacency parameter matrix* $\mathbf{A} \in \mathbb{R}^{n \times n}$. *Given a set of cascades* $\mathcal{C}$, *network inference seeks to estimate the adjacency parameter matrix* $\mathbf{A}$ *based on* $\mathcal{C}$, *thereby inferring the underlying edge set* $\mathcal{E}$ *of* $\mathbf{G}$.

**DEFINITION 2 (INFLUENCE ESTIMATION).** *Consider a diffusion network* $\mathbf{G} = (\mathcal{V}, \mathcal{E})$ *with node set* $\mathcal{V}$, *edge set* $\mathcal{E}$, *and the associated adjacency parameter matrix* $\mathbf{A}$. *Given a set of seed nodes* $\mathcal{S} \subset \mathcal{V}$ *and a time window* $T$, *influence estimation aims to estimate the expected number of nodes influenced by* $\mathcal{S}$ *within the time window* $T$ *under the continuous-time independent cascade (CIC) diffusion model.*

## 3 DIFFUSION FRAMEWORK

In this section, we develop the framework to model *continuous-time diffusions* in a diffusion medium.

### 3.1 Conditional Diffusion Rate

During one diffusion process, an inactive node can have multiple active incoming neighbors, thus possibly being influenced by them simultaneously. The parameterization of the cumulative impact conditioned on a given observation is determined by the joint diffusion rates of its active neighbors. In this context, $\gamma_t \in \mathbb{R}^n$ denotes the *conditional diffusion rate*, for which the following holds.

**LEMMA 1.** *Given an observation* $\mathbf{h}_t$ *with* $\phi_{t,v} = 0$ *for node* $v \in \mathcal{V}$, *we have*

$$\gamma_{t,v} = \sum_{u \in \mathcal{N}_v} (\lambda_{uv} \cdot \phi_{t,u}). \tag{1}$$

Let $\Gamma_t$ be the *expected* conditional diffusion rate over the randomness of observations at time $t$, i.e., $\Gamma_t = \sum_{\mathbf{h}_t \sim \mathcal{C}} \gamma_t \cdot p(\mathbf{h}_t)$ where $p(\mathbf{h}_t)$ is the probability of observation $\mathbf{h}_t$. By Equation (1), we have

**LEMMA 2.** *Given a random observation* $\mathcal{H}_t$, *we have*

$$\Gamma_t = \mathbb{E}[(\mathbb{1} - \Phi_t) \odot \mathbf{A}^\top \Phi_t \mid \mathcal{H}_t], \tag{2}$$

*where* $\mathbb{1} = \{1\}^n$ *is an n-dimension all-ones vector and* $\odot$ *is the Hadamard product.*

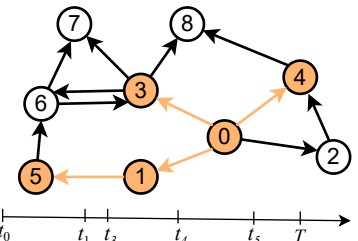

**Figure 2: Illustration of a diffusion cascade.**

### 3.2 Continuous-time Diffusion Propagation

As described in Section 2.2, influence from source sets spreads along edges to other nodes as a continuous-time cascade. Seeing such influence propagations as temporal evolving dynamics, a continuous-time diffusion among nodes of a network is essentially a *continuous-time dynamical system* (CDS), succinctly defined as follows.

**DEFINITION 3 (CONTINUOUS-TIME DYNAMICAL SYSTEM [4, 22]).** *A continuous-time dynamical system consists of a phase space* $\mathcal{X}$ *and a transformation map* $\Omega : (t, \mathcal{X}) \to \mathcal{X}$ *where* $t \in \mathbb{R}_0^+$ *is the time.*

Accordingly, we apply CDS to diffusion networks and formalize the concept of *continuous-time diffusion propagation* (CDP).

**DEFINITION 4 (CONTINUOUS-TIME DIFFUSION PROPAGATION).** *Consider a diffusion network* $\mathbf{G} = (\mathcal{V}, \mathcal{E})$ *associated with adjacency parameter matrix* $\mathbf{A}$, *source set* $\mathcal{S} \subseteq \mathcal{V}$, *and time window* $T$. *The continuous-time diffusion propagation starting from* $\mathcal{S}$ *by time* $T$ *is defined as a tuple* $(\mathbf{G}, \mathcal{X}_T, \Omega)$, *where* $\mathcal{X}_T$ *is the state space of* $\mathcal{V}$ *and* $\Omega$ *is the state transition function, which advances a state* $\Phi_t \in \mathcal{X}_T$ *to* $\Phi_{t+\tau}$ *for an infinitesimal interval* $\tau \to 0^+$ *with respect to graph* $\mathbf{G}$.

The foundation of CDP is the state transition function $\Omega$, which quantifies the evolution of node states over time. In particular, function $\Omega$ computes node states in the future time $t + \tau$ solely upon states at the current time $t$. To capture the transition within $\tau$, we first establish the *ordinary differential equation* (ODE) of $\Omega$. Consequently, CDP can be formulated as follows.

**THEOREM 1.** *Let* $\mathcal{S}$ *be a given source set and* $\mathcal{H}_t$ *be the random observation variable at time* $t$. *Consider the infinitesimal interval* $\tau \to 0^+$. *The continuous-time diffusion propagation for all* $t \geq 0$ *is defined as*

$$\mathbb{E}[\Phi_{t+\tau} - \Phi_t \mid \mathcal{H}_t] = \tau \Gamma_t, \tag{3}$$

$$\Gamma_t = \mathbb{E}[(\mathbb{1} - \Phi_t) \odot \mathbf{A}^\top \Phi_t \mid \mathcal{H}_t], \tag{4}$$

$$\Phi_0 = \mathbb{1}_{\mathcal{S}}, \tag{5}$$

*where* $\odot$ *denotes the Hadamard product and* $\mathbb{1}_{\mathcal{S}} \in \{0, 1\}^n$ *is the indicator vector, with* $\mathbb{1}_v = 1$ *for* $v \in \mathcal{S}$ *and* $\mathbb{1}_u = 0$ *otherwise.*

### 3.3 Framework of Continuous-time Diffusion

Note that $\Gamma_t$ and $\mathbb{E}[\Phi_t]$ at time $t = 0$ can be simply known from Equation (4) and Equation (5) respectively, once the source set $\mathcal{S}$ is given. Furthermore, Equation (3) shows that the exact value of $\mathbb{E}[\Phi_{t+\tau}]$ within interval $\tau \to 0^+$ is derived from $\mathbb{E}[\Phi_t]$. Instead, by relaxing the constraint from $\tau \to 0^+$ to $\varepsilon > 0$ for a small interval $\varepsilon$, we are able to acquire an approximation of the future state $\mathbb{E}[\Phi_{t+\varepsilon}]$.

For this, we resort to approximate $\mathbb{E}[\Phi_{t+\varepsilon}]$ based upon $\mathbb{E}[\Phi_t]$ and $\Gamma_t$, after which we then estimate $\Gamma_{t+\varepsilon}$, i.e.,

$$\mathbb{E}[\Phi_{t+\varepsilon} \mid \mathcal{H}_t] \approx \mathbb{E}[\Phi_t \mid \mathcal{H}_t] + \varepsilon\Gamma_t, \tag{6}$$

$$\Gamma_{t+\varepsilon} \approx \mathbb{E}[(\mathbb{1} - \Phi_{t+\varepsilon}) \odot \mathbf{A}^\top \Phi_{t+\varepsilon} \mid \mathcal{H}_t]. \tag{7}$$

Based on this analysis, we propose our framework FIM (**F**ramework of d**I**ffusion approxi**M**ation) to approximate $\{\mathbb{E}[\Phi_\varepsilon], \mathbb{E}[\Phi_{2\varepsilon}], \dots, \mathbb{E}[\Phi_T]\}$ progressively.

Approximation by $\varepsilon$ instead of $\tau$ inevitably incurs an approximation error at each iteration. Let $\xi(t, \varepsilon) \in \mathbb{R}^n$ be the corresponding approximation error of $\mathbb{E}[\Phi_{t+\varepsilon}]$ given observation $\mathcal{H}_t$, i.e., $\xi(t, \varepsilon) = \mathbb{E}[\Phi_t - \Phi_{t+\varepsilon} \mid \mathcal{H}_t] + \varepsilon\Gamma_t$. To quantify $\xi(t, \varepsilon)$, we derive the following theorem.

THEOREM 2. *Given time $t \in [0, T)$ and time interval $\varepsilon \in (0, T - t]$, the approximation error $\xi(t, \varepsilon)$ at $t$ is*

$$\xi(t, \varepsilon) = \varepsilon\Gamma_t + \exp(-\varepsilon\Gamma_t) - \mathbb{1}. \tag{8}$$

As shown, $\xi(t, \varepsilon)$ is therefore monotonically increasing when $\varepsilon \geq 0$ and $\xi(t, \varepsilon) = 0$ if $\varepsilon = 0$.

**Complexity.** The time complexity is dominated by the calculation of $\Gamma$ in Equation (4), with a cost of $O(m)$, where $m$ is the number of non-zero elements in $\mathbf{A}$. Thus the total time complexity of the approximation is $O(\frac{T}{\varepsilon}(m + n))$.

## 4 NETWORK INFERENCE AND INFLUENCE ESTIMATION

FIM assumes that the adjacency parameter matrix $\mathbf{A}$ is known. However, this assumption rarely holds, as $\mathbf{A}$ is typically unknown in reality. In this section, we aim to learn the parameter matrix $\mathbf{A}$ by leveraging FIM to model the cascade data and then infer the underlying network structure. Afterwards, we develop a sampling technique for continuous-time influence estimation.

### 4.1 Network Inference

As introduced in Section 2.2, a cascade $\mathbf{c}$ within time $T$ records the activation time $t_v$ of each node $v \in \mathcal{V}$. Given cascade $\mathbf{c}$, we can restore its diffusion process by generating a series of diffusion observations $\mathbf{h}_t$ and node states $\phi_t$. In particular, given a small interval $\varepsilon$, we construct $\mathbf{h}_{k\varepsilon}$ and $\phi_{k\varepsilon}$ for $k \in \{0, 1, \cdots, \lfloor T/\varepsilon \rfloor\}$ by comparing $t_v$ with $k\varepsilon$ for each node $v$. Specifically, we set $\mathbf{h}_{k\varepsilon,v} = t_v$ and $\phi_{k\varepsilon,v} = 1$ if $t_v \leq k\varepsilon$, else $\mathbf{h}_{k\varepsilon,v} = \infty$ and $\phi_{k\varepsilon,v} = 0$ for $\forall v \in \mathcal{V}$.

With observation $\mathbf{h}_{(k-1)\varepsilon}$ and state $\phi_{(k-1)\varepsilon}$ at $t = (k-1)\varepsilon$, the expected state $\mathbb{E}[\Phi_{k\varepsilon} \mid \mathbf{h}_{(k-1)\varepsilon}]$ conditioned on $\mathbf{h}_{(k-1)\varepsilon}$ can be estimated by FIM, after which the corresponding *binary cross-entropy* (BCE) loss $\ell_{k\varepsilon}$ over all nodes is calculated as

$$\ell_{k\varepsilon} = \sum_{v\in\mathcal{V}} \phi_{k\varepsilon,v} \log \mathbb{E}[\Phi_{k\varepsilon,v} \mid \mathbf{h}_{(k-1)\varepsilon}]$$
$$+ (1 - \phi_{k\varepsilon,v}) \log(1 - \mathbb{E}[\Phi_{k\varepsilon,v} \mid \mathbf{h}_{(k-1)\varepsilon}]). \tag{9}$$

By simulating the complete cascade $\mathbf{c}$, the total loss $\ell(\mathbf{c})$ of $\mathbf{c}$ is $\ell(\mathbf{c}) = \sum_{k=1}^{\lfloor T/\varepsilon \rfloor} \ell_{k\varepsilon} + \ell_T$. Formally, the procedure of FIM simulating cascade $\mathbf{c}$ is presented in Algorithm 1.

**Training of parameter matrix A.** The training process of $\mathbf{A}$ is given in Algorithm 2. Specifically, we are first given a set $\mathcal{V}$ with $n$ nodes, the time interval $\varepsilon$, and a set of cascades $C$. Next, the adjacency parameter matrix $\mathbf{A}$ is initialized by random sampling from

---

**Algorithm 1:** Diffusion approximation by FIM

**Input:** Cascade batch $C_B$, time interval $\varepsilon$, parameter matrix $\mathbf{A}$
**Output:** Loss $\ell(C_B)$

**1** **for** *cascade* $\mathbf{c} \in C_B$ **do**
**2**      Initialize observation $\mathbf{h}_0$ and state $\phi_0$ according to $\mathbf{c}$;
**3**      **for** $k \leftarrow 1$ *to* $\lfloor T/\varepsilon \rfloor$ **do**
**4**          $\gamma_{(k-1)\varepsilon} \leftarrow (\mathbb{1} - \phi_{(k-1)\varepsilon}) \odot \mathbf{A}^\top \phi_{(k-1)\varepsilon}$;
**5**          $\mathbb{E}[\Phi_{k\varepsilon} \mid \mathbf{h}_{(k-1)\varepsilon}] \leftarrow \phi_{(k-1)\varepsilon} + \varepsilon\gamma_{(k-1)\varepsilon}$;
**6**          Obtain $\mathbf{h}_{k\varepsilon}$ and state $\phi_{k\varepsilon}$ from $\mathbf{c}$;
**7**          Calculate $\ell_{k\varepsilon}$ as Equation (9);
**8**      $\varepsilon \leftarrow T - k\varepsilon$;
**9**      Calculate $\ell_T$ conditioned on $\ell_{\lfloor T/\varepsilon \rfloor \varepsilon}$ by following the procedure from Line 4 to Line 7;
**10**      $\ell(\mathbf{c}) = \sum_{k=1}^{\lfloor T/\varepsilon \rfloor} \ell_{k\varepsilon} + \ell_T$;
**11** $\ell(C_B) \leftarrow \frac{1}{|C_B|} \sum_{\mathbf{c}\in C_B} \ell(\mathbf{c})$;
**12** **return** $\ell(C_B)$;

---

**Algorithm 2:** Training of parameter matrix $\mathbf{A}$

**Input:** Node set $\mathcal{V}$ with $n = |\mathcal{V}|$, cascade set $C$, time interval $\varepsilon$, batch size $B$
**Output:** Estimated parameter matrix $\mathbf{A}$

**1** Initialize parameter matrix $\mathbf{A}$ using Gaussian distribution $\mathcal{N}(0, \frac{1}{n^2})$;
**2** **for** $k \leftarrow 1$ *to* $\lceil |C|/B \rceil$ **do**
**3**      Sample a batch of cascades $C_B$ from $C$;
**4**      Calculate loss $\ell(C_B)$ by Algorithm 1 with inputs $C_B$, $\varepsilon$, and $\mathbf{A}$;
**5**      Apply SGD to $\ell(C_B)$ and update $\mathbf{A}$ during backpropagation;
**6** **return** $\mathbf{A}$;

---

a Gaussian distribution with $\mu = 0$ and $\sigma = \frac{1}{n}$. During training, we sample a batch $C_B$ of cascades from $C$ and compute the average loss $\ell(C_B)$ using Algorithm 1. Subsequently, we apply stochastic gradient descent (SGD) on the loss $\ell(C_B)$ to update $A$ during backpropagation. When a number of cascades of the order of $|C|$ have been sampled during training, the resulting matrix $\mathbf{A}$ is returned.

Once the parameter matrix $\mathbf{A}$ is estimated, the underlying network structure can be inferred accordingly. By following the literature [4, 10, 11, 13], an empirical threshold $\lambda^*$ can be set such that an edge $\langle u, v \rangle$ exists if $\mathbf{A}[u, v] \geq \lambda^*$. (More details in Appendix A.2).

### 4.2 Influence Estimation

Given a graph $\mathbf{G} = (\mathcal{V}, \mathcal{E})$ with the adjacency parameter matrix $\mathbf{A}$, a source set $\mathcal{S} \subseteq \mathcal{V}$, and a time window $T$, let $I(T, \mathcal{S})$ be the number of activated nodes in a cascade $c$ sampled from $C(T, \mathcal{S})$. The influence estimation problem is to compute the expected number of nodes $\mathbb{E}[I(T, \mathcal{S})]$ influenced by $\mathcal{S}$ within the time window $T$.

It is intuitive that a node $u$ is influenced by the source node $s \in \mathcal{S}$ that has the shortest time (path) to $u$ under the CIC model, where the edges are weighted by their transmission times, which is known as the *shortest path property* in the literature [9, 14, 16]. By consequence, one instance of $I(T, \mathcal{S})$ can be calculated as follows. By sampling the diffusion time $t_{u,v}$ of each edge $\langle u, v \rangle \in \mathcal{E}$ according to the diffusion function $p(\lambda_{uv}, t)$, $I(T, \mathcal{S})$ is the number of nodes reachable by $\mathcal{S}$ within time $T$. Existing methods [9, 16] first identify one activation set for each source node $s \in \mathcal{S}$ and then take the union over all the source nodes to calculate $I(T, \mathcal{S})$.

---

**Algorithm 3:** Influence Estimation

**Input:** Graph $\mathbf{G}$, matrix $\mathbf{A}$, set $\mathcal{S}$, time window $T$,
      parameters $\eta, \delta$

**Output:** Estimation of $\mathbb{E}[I(T, \mathcal{S})]$

1   $u \leftarrow$ randomly select from $\mathcal{S}$;

2   add edges $\langle u, v \rangle \ \forall v \in \mathcal{S} \setminus \{u\}$ to $\mathbf{G}$;

3   Set $t_{u,v} \leftarrow 0$ for all added edges;

4   $\theta \leftarrow \lceil \frac{1}{2\eta^2} \ln \frac{2}{\delta} \rceil$, $I_{T,\mathcal{S}} \leftarrow 0$;

5   **for** $i \leftarrow 1$ *to* $\theta$ **do**

6      Run Dijkstra's algorithm from $u$;

7      Sample $t_e$ from $p(\lambda_e, t)$ for newly visited edge $e$;

8      Terminate if diffusion time exceeds $T$;

9      $I_{T,\mathcal{S}} \leftarrow I_{T,\mathcal{S}} + |\{v \mid \mathrm{D}(v, \mathcal{S}) \leq T\}|$;

10   **return** $\frac{I_{T,\mathcal{S}}}{\theta}$;

---

However, this approach will inevitably incur an unnecessary computation overhead, since one node can be visited by multiple source nodes. To avoid this issue, we formalize the concept of the *shortest diffusion time of set* (SDTS), by which we are able to estimate the expected influence $\mathbb{E}[I(T, \mathcal{S})]$ directly, without inspecting the individual influence of each source node.

DEFINITION 5 (SHORTEST DIFFUSION TIME OF SET). *Given a graph* $\mathbf{G} = (\mathcal{V}, \mathcal{E})$, *matrix* $\mathbf{A}$, *and a source set* $\mathcal{S} \subseteq \mathcal{V}$, *the transmission time* $t_{u,v}$ *of edge* $\langle u, v \rangle \in \mathcal{E}$ *is sampled from the function* $p(\lambda_{uv}, t)$. *The shortest diffusion time* $\mathrm{D}(u, \mathcal{S})$ *of the set* $\mathcal{S}$ *from a node* $u \in \mathcal{V}$ *is* $\mathrm{D}(u, \mathcal{S}) = \min\{d(v, u) \mid \forall v \in \mathcal{S}\}$ *where* $d(v, u)$ *is the shortest diffusion time from* $v$ *to* $u$, *i.e., the shortest distance in the induced graph with edges weighted by transmission times.*

We call such a graph $\mathcal{G}$ obtained from $\mathbf{G}$ by sampling a $t_{u,v}$ value for each edge $\langle u, v \rangle \in \mathcal{E}$ an *instance* of $\mathbf{G}$. In such an instance, the corresponding influence $I(T, S) = |\{u \mid \mathrm{D}(u, \mathcal{S}) \leq T\}|$ is the number of nodes reachable by $\mathcal{S}$ within time $T$. Hence, the expected influence $\mathbb{E}[I(T, \mathcal{S})]$ is formulated as $\mathbb{E}[I(T, \mathcal{S})] = \mathbb{E}_{t_{u,v} \sim p(\lambda_{uv}, t)}\big[|\{u \mid \mathrm{D}(u, \mathcal{S}) \leq T\}|\big]$. According to the Hoeffding Inequality [20], we have the following lemma.

LEMMA 3. *By generating a number of* $\theta = \frac{1}{2\eta^2} \ln \frac{2}{\delta}$ *instances* $\mathcal{G}$ *with* $\eta, \delta \in (0, 1)$ *to estimate* $I(T, \mathcal{S})$, *we have*

$$\Pr\left[\left|\frac{\sum_{i=1}^{\theta} I_i(T, \mathcal{S})}{\theta} - \mathbb{E}[I(T, \mathcal{S})]\right| \leq \eta n\right] \geq 1 - \delta. \tag{10}$$

Note that graph instances $\mathcal{G}$ based on a source set $\mathcal{S}$ can be efficiently constructed by leveraging a variant of *Dijkstra's algorithm* [7]. First, we randomly select a node $u \in \mathcal{S}$ as the source. Then, we add extra edges $\langle u, v \rangle$ to $\mathbf{G}$ for all $v \in \mathcal{S} \setminus \{u\}$ if $\langle u, v \rangle$ does not exist and set $t_{u,v} = 0$. Finally, we run Dijkstra's algorithm on the augmented $\mathbf{G}$, starting from $u$ (i.e., the variant for all shortest paths from $u$). During the traversal (a diffusion simulation), we sample $t_e$ for each visited edge $e \in \mathcal{E}$ as aforementioned. A traversal terminates when the current diffusion time exceeds $T$. The pseudo-code for this approach is presented in Algorithm 3.

Based on a training set of cascades, the expected influence $\mathbb{E}[I(T, \mathcal{S})]$ of any given set $\mathcal{S}$ is obtained from the inferred network by FIM. We quantify the effect of network inference errors on $\mathbb{E}[I(T, \mathcal{S})]$ in the following theorem.

THEOREM 3. *Let* $\rho^* \in \mathbb{R}^m$ *be the vector of true diffusion rates associated with* $m$ *edges and let* $\rho$ *be the estimation by FIM. For any given seed set* $\mathcal{S} \subseteq \mathcal{V}$, *let* $T^*$ *and* $T$ *denote time windows for the* $\rho^*$ *(true) and* $\rho$ *(inferred) settings, respectively. Assume that* $\|\rho^* - \rho\|_\infty \leq \epsilon$ *and* $\min(\rho^*) = c\epsilon$ *for a constant* $c > 1$ *where* $\min(\rho^*)$ *is the minimal element in* $\rho^*$ *and* $\epsilon \geq 0$. *To cover the same nodes as with* $(\rho^*, T^*)$, *we have* $\mathbb{E}[T] \in \left( \frac{1 - \exp(-\epsilon T^*)}{\epsilon}, \frac{cT^*}{c-1} \right)$.

**Complexity.** The cost of Dijkstra's algorithm is $\Theta(m + n \log n)$, safely ignoring the additional $|\mathcal{S}|$ edges in this asymptotic formulation. Moreover, it is unlikely that all $m + |\mathcal{S}|$ edges and $n$ nodes are visited since the diffusion terminates early, once the time limit $T$ is reached. Therefore, the worst-case time complexity of Algorithm 3 is $O\big((m + n \log n) \frac{1}{\eta^2} \ln \frac{1}{\delta}\big)$.

**Influence Maximization.** When the network topology and parameters $\mathbf{A}$ are known, continuous-time influence maximization (CIM) can be addressed by existing algorithms, e.g., [37]. Therefore, CIM is out of the scope of our paper. Nevertheless, we asses in one experiment the performance of parameter learning approaches (including ours), by evaluating the spread performance of the downstream CIM task, with [37] over learned matrix $\mathbf{A}$. More details on the rationale and results of this experiment are provided next.

## 5 EXPERIMENTS

**Datasets and experimental setup.** We collected 3 real-world cascade datasets, *MemeTracker* (Meme) [26], *Weibo* [41], and *Twitter* [19] from diverse applications (all publicly available).

By following the literature [8, 13, 16, 18], we exploit the *Kronecker network model* [27] to also generate three types of synthetic networks: Hierarchical (Hier) [5], Core-periphery (Core) [28], and Random (Rand) networks, with parameter matrices $[0.9, 0.1; 0.1, 0.9]$, $[0.9, 0.5; 0.5, 0.3]$, and $[0.5, 0.5; 0.5, 0.5]$, respectively. For each type, we generated two networks, consisting of 1024 and 2048 nodes, with edge counts being 4 times the node counts, denoted as $\text{Hier}_{1024}$ ($\text{Hier}_{2048}$), $\text{Core}_{1024}$ ($\text{Core}_{2048}$), and $\text{Rand}_{1024}$ ($\text{Rand}_{2048}$) respectively. For the diffusion rate $\lambda_{uv}$, we sample each $\lambda_{uv}$ uniformly from $(0, 0.1)$. Furthermore, we generate $20,000$ or $10,000$ continuous-time cascades for each synthetic network.

Finally, for a fair comparison, the HR synthetic dataset, which was extensively tested in [18], is also included in our experiments. The statistics of the 10 datasets are summarized in Table 2, while a detailed description of the real-world datasets and their associated cascades is given in Appendix A.2.1.

### 5.1 Parameter learning and network inference

**Baselines and setting.** On the 10 datasets, our objective was to compare FIM with two baselines, the well-known NetRate [11] and the recent state-of-the-art NMF [18], with both implementations being the official ones published by the authors. We varied the size of the time window $T \in \{5, 8, 10, 12, 15\}$ by fixing $\epsilon = 1.0$ and we varied $\epsilon \in \{0.5, 1.0, 1.5, 2.0\}$ by fixing $T = 10$. We randomly split the cascades into training / validation / testing sets with percentages 80% / 10% / 10%, respectively.

## Table 2: Dataset details.

| Dataset | HR | Hier$_{1024}$ / Hier$_{2048}$ | Core$_{1024}$ / Core$_{2048}$ | Rand$_{1024}$ / Rand$_{2048}$ | MemeTracker | Weibo | Twitter |
|---|---|---|---|---|---|---|---|
| #Nodes | 128 | 1024 / 2048 | 1024 / 2048 | 1024 / 2048 | 498 | 8,190 | 12,677 |
| #Cascades | 10,000 | 20,000 / 10,000 | 20,000 / 10,000 | 20,000 / 10,000 | 8,304 | 43,365 | 3,461 |

### Table 3: BCE loss on Meme and HR.

| | Method | Time window T (s) | | | | | Time interval $\varepsilon$ | | | |
|---|---|---|---|---|---|---|---|---|---|---|
| | | 5 | 8 | 10 | 12 | 15 | 0.5 | 1.0 | 1.5 | 2.0 |
| HR | FIM | 0.26 | 0.73 | 1.20 | 1.79 | 2.89 | 2.30 | 1.20 | 0.67 | 0.67 |
| | NMF | 0.30 | 0.95 | 1.61 | 2.35 | 3.07 | 3.08 | 1.61 | 0.88 | 0.85 |
| | NetRate | 0.42 | 1.16 | 1.91 | 2.88 | 4.70 | 3.49 | 1.91 | 1.14 | 1.23 |
| Meme | FIM | 0.44 | 0.68 | 0.84 | 1.00 | 1.25 | 1.63 | 0.84 | 0.53 | 0.46 |
| | NMF | 0.51 | 0.71 | 0.88 | 1.03 | 1.26 | 1.66 | 0.88 | 0.56 | 0.48 |

### Table 4: Influence maximization on Meme and Core$_{2048}$.

| Dataset | Method | Seed Set Size $|\mathcal{S}|$ | | | | | | |
|---|---|---|---|---|---|---|---|---|
| | | 4 | 5 | 6 | 7 | 8 | 9 | 10 |
| Meme | FIM | 65.85 | 75.18 | 88.33 | 108.88 | 113.90 | 127.11 | 136.67 |
| | NMF | 53.14 | 64.78 | 70.66 | 76.10 | 83.39 | 88.23 | 94.95 |
| Core$_{2048}$ | FIM | 517.12 | 576.19 | 559.19 | 634.83 | 636.26 | 639.38 | 723.88 |
| | NMF | 444.25 | 506.17 | 476.44 | 570.90 | 591.80 | 637.40 | 567.28 |

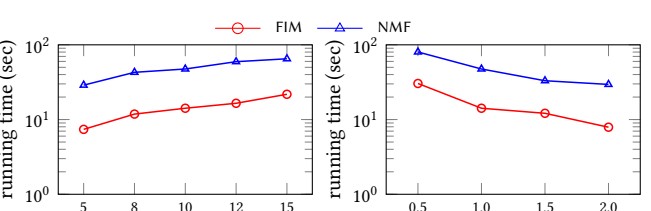

Figure 3: Running time on HR.

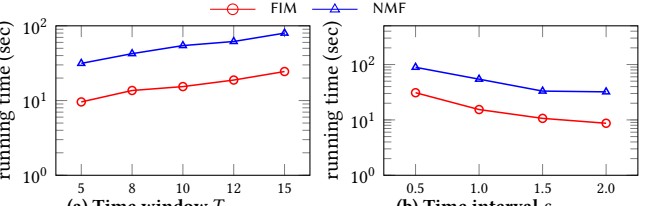

Figure 4: Running time on Meme.

**Limitations of the baseline methods.** We start by reporting that, on our reasonably powerful server[1], both NMF and NetRate are *out of memory* (OOM) on the larger datasets (Weibo and Twitter) and thus their performance is not available on them. Furthermore, NetRate could not finish in less than 12 hours on MemeTracker and the synthetic datasets, except the smallest one (HR). We believe this is due to the joint effect of the node count, cascade count, and cascade sizes. We can therefore conclude from these outcomes that NetRate– especially with the official Matlab implementation – is not efficient for large networks[2].

**Parameter learning on HR and MemeTracker.** In a first experiment, we compare FIM's parameter learning performance on (i) the only dataset common to all three methods (HR), and (ii) the only real-world dataset common to both NMF and FIM (MemeTracker). (We defer a similar comparison between FIM and NMF, on the other six synthetic datasets, to the next experiment, where they are complemented by effectiveness scores on network inference based on the inferred parameters.) Table 3 presents the BCE loss results – as defined in Equation (9) – for parameter learning from cascades with a varying time window $T$ and time interval $\varepsilon$. We can notice that FIM outperforms NMF by achieving slightly smaller BCE loss on both datasets. Furthermore, FIM consistently achieves notably smaller BCE loss compared to NetRate on HR, across all settings. These findings provide good evidence of FIM's superior ability to learn a high-quality parameter matrix **A**.

Furthermore, Figures 3 and 4 plot the training time of FIM and NMF, with varying time window values $T$ and time intervals $\varepsilon$, on HR and MemeTracker respectively. As displayed, FIM runs notably

faster than NMF, with speedups up to 3× ∼ 4×. Moreover, the efficiency advantage becomes more obvious as the data grows.

**CIM based evaluation on the learned parameter matrix.** Following the previous experiment, we further assess of quality of each inferred matrix **A** from the perspective of *continuous influence maximization (CIM)*, based on *test cascades* (unseen during training), as follows.

Recall that CIM algorithms are able to identify the seed set $\mathcal{S}$ with the largest expected spread $\mathbb{E}[\mathcal{S}]$ in a given diffusion network (as per **A**). In our setting, following the convention [8, 18], we can obtain the *ground-truth* spread (i.e., quality) of a source set $\mathcal{S}$ as follows: randomly sample from the test cascades one for each node in $\mathcal{S}$, and take the union of the nodes from the sampled cascades, leading to a single estimate of $\mathbb{E}[\mathcal{S}]$; repeat this process 1000 times to obtain the average over estimates as $\mathcal{S}$'s ground-truth spread. Therefore, generically, when a state-of-the-art CIM algorithm selects a seed set $\mathcal{S}$ in the network defined by an estimated parameter matrix **A**, we can assess the quality of $\mathcal{S}$ and, by design, also the accuracy of **A** itself. *A larger ground-truth spread means a higher estimation quality for the parameter matrix* **A**. (A more detailed justification for this CIM experiment can be found in Appendix A.2.2.)

As the HR dataset is unsuitable for this CIM experiment, we used Core$_{2048}$ instead. This is because HR's cascades contain multiple source nodes (see the dataset's description in Appendix A.2.1), hence ground-truth spread cannot be obtained as described. Nevertheless, we describe in Appendix A.3 an influence estimation (IE) experiment on HR, designed for multi-source cascades, to assess the quality of the estimated **A** by FIM, NMF, and NetRate.

We applied the state-of-the-art CIM algorithm [37] on the estimated **A** of both MemeTracker and Core$_{2048}$, selecting seed sets $\mathcal{S}$ with sizes from $|\mathcal{S}| = 4$ to $|\mathcal{S}| = 10$. Table 4 shows the spread

[1]We refer to Appendix A.2.2 for the details on the server's configurations.
[2]We stress that, as a sanity check for our code and computing environment, we reproduced the running times of NetRate from the paper [11], as shown in Appendix A.2

**Table 5: Results on the 6 synthetic datasets**

|  | Method | $\text{Rand}_{1024}$ | $\text{Hier}_{1024}$ | $\text{Core}_{1024}$ | $\text{Rand}_{2048}$ | $\text{Hier}_{2048}$ | $\text{Core}_{2048}$ |
|---|---|---|---|---|---|---|---|
| BCE loss | FIM | 0.08 | 0.04 | 0.17 | 0.06 | 0.04 | 0.14 |
|  | NMF | 0.11 | 0.06 | 0.20 | 0.08 | 0.06 | 0.19 |
| $F_1$-score | FIM | 0.60 | 0.75 | 0.58 | 0.41 | 0.42 | 0.34 |
|  | NMF | 0.44 | 0.49 | 0.36 | 0.32 | 0.36 | 0.16 |

**Table 6: BCE loss on Weibo and Twitter.**

|  | Method | Time window T (s) | | | | | Time interval $\varepsilon$ | | | |
|---|---|---|---|---|---|---|---|---|---|---|
|  |  | 5 | 8 | 10 | 12 | 15 | 0.5 | 1.0 | 1.5 | 2.0 |
| Weibo | FIM | 0.69 | 1.07 | 1.79 | 1.93 | 3.25 | 2.65 | 1.79 | 0.82 | 0.77 |
| Twitter | FIM | 0.003 | 0.006 | 0.008 | 0.01 | 0.014 | 0.015 | 0.008 | 0.005 | 0.005 |

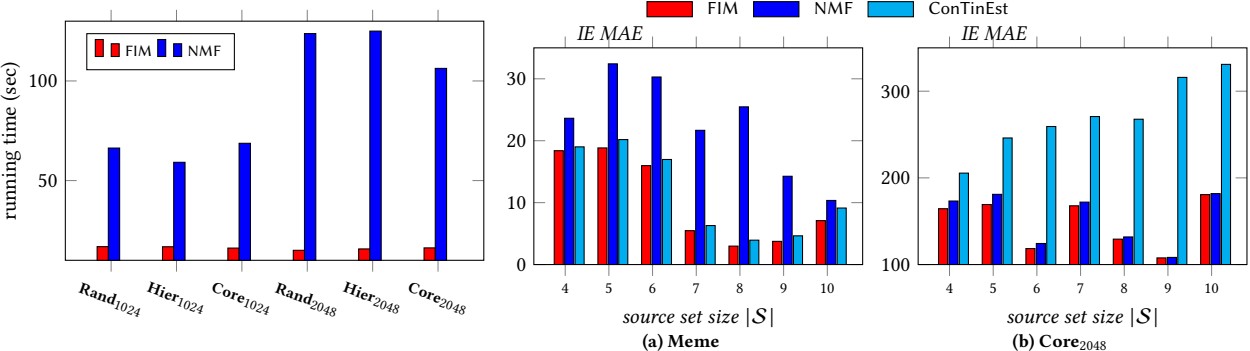

**Figure 5: Running time on synthetic datasets.**

**Figure 6: Influence estimation MAE.**

results for seed sets selected based on the estimated $\mathbf{A}$ of these two datasets, by FIM and NMF respectively. We can observe that the spread values obtained based on FIM's estimated $\mathbf{A}$ are significantly larger than those based on NMF's output. This observation further supports the high learning ability of FIM over NMF.

**Network inference on synthetic networks.** Recall that, once the parameter matrix is learned, we can infer a most likely diffusion topology by applying a predefined threshold $\lambda^*$ to the estimated $\mathbf{A}$, i.e., $\mathbf{A}[u, v] \geq \lambda^*$ indicating the existence of the edge $\langle u, v \rangle$. In this experiment, we assess the network inference performance of FIM and NMF, on the 6 synthetic datasets, for which the ground-truth networks are available. Here, we fixed the time window $T = 10$ and the time interval $\varepsilon = 1.0$. When an estimated parameter matrix $\mathbf{A}$ is obtained, we set the threshold $\lambda^* = 0.01$ to determine the existence of an edge.

We use the *F1-score* metric to measure the performance on predicting the existence of edges, and the results are presented in Table 5. We can observe that the F1-scores of FIM are significantly higher compared to those of NMF, especially on $\text{Hier}_{1024}$ and $\text{Core}_{1024}$. For a complete perspective, besides the F1-score, we also provide the BCE loss of parameter learning on those synthetic datasets. Once again, FIM achieves consistently a smaller BCE loss than NMF, for all the synthetic datasets.

Finally, in this same setting ($T = 10$ and $\varepsilon = 1.0$), Figure 5 compares the training time of FIM and NMF on the synthetic datasets. Similar to the results of Figures 3 and 4, FIM outperforms NMF significantly (e.g., speedup of up to 8.24× on $\text{Hier}_{2048}$).

### 5.2 Influence estimation on inferred networks

**Setting and baselines.** By fixing the underlying inferred networks, we can evaluate the performance of FIM (Section 4.2) for influence estimation (IE), against the baselines ConTinEst [9] and NMF [18]. We randomly generate a series of source sets $\mathcal{S}$ with size $|\mathcal{S}| \in \{4, 5, 6, 7, 8, 9, 10\}$ from the test cascades. We then compute the *mean absolute error* (MAE) between the spread estimated by each method

and the ground-truth, denoted by *IE MAE*, as well as the influence estimation time. For a fair comparison, we use the same sample size for FIM and ConTinEst (no sampling in NMF).

**Results.** Due to space limitations, we chose to present the results of IE on the inferred networks from MemeTracker and $\text{Core}_{2048}$, since we observed similar results across the other synthetic datasets. From the results on IE MA (Figure 6) we can conclude FIM achieves the smallest MAE on both datasets, regardless of the source set size. The IE MAEs of NMF are notably larger than those of the competitors on MemeTracker, while ConTinEst performs clearly the worst on $\text{Core}_{2048}$. This observation also indicates the robustness of FIM across diverse datasets.

Figure 7 compares the influence estimation times for the three tested methods. We can observe that FIM is *orders of magnitude* faster than NMF and ConTinEst on both datasets, which confirms the gains by our proposed sampling technique SDTS (Section 4.2). In particular for NMF, its running time consists of the model loading time, i.e., loading the pre-trained model for IE, and the estimation time, while the former obviously dominates the latter, as observed in our experiment.

### 5.3 Scalability evaluation on large networks

**Larger real-world datasets.** To further explore the scalability of FIM, we employ two large real-world datasets, i.e., Weibo and Twitter, containing 8, 190 and 12, 677 nodes respectively, for continuous-time network inference. To the best of our knowledge, Twitter is the largest real-world dataset tested in the literature for this problem.

Table 6 presents the BCE loss under various time window values $T$ and time intervals $\varepsilon$, while Figure 8 plots the corresponding running time of FIM. It is worth pointing out that the BCE loss of FIM on Twitter is surprisingly low (at most 0.01). Regarding the running time on the two datasets, FIM completes the training phase within 30 seconds for the largest time window $T = 15$ and around 35 seconds for the smallest $\varepsilon = 0.5$. These results not only validate the practical interest of FIM for network inference from cascades,



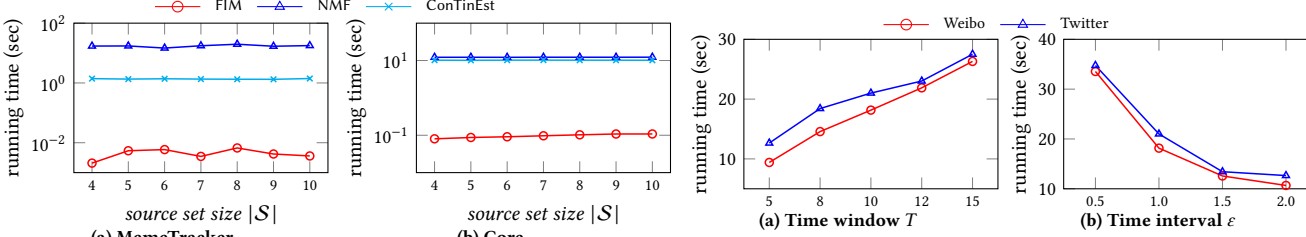

(a) MemeTracker

(b) Core$_{2048}$

**Figure 7: Influence estimation times.**

**Table 7: IE MAE and running time (RT) on Core$_{4096}$.**

| | Method | Seed Set Size $|\mathcal{S}|$ | | | | | | |
|---|---|---|---|---|---|---|---|---|
| | | 4 | 5 | 6 | 7 | 8 | 9 | 10 |
| IE MAE | FIM | 348.62 | 371.58 | 355.36 | 253.58 | 299.41 | 290.72 | 300.26 |
| | ConTinEst | 401.92 | 433.66 | 402.42 | 384.06 | 406.90 | 362.82 | 353.24 |
| RT (sec) | FIM | 19.22 | 20.87 | 21.42 | 19.13 | 20.19 | 20.98 | 19.93 |
| | ConTinEst | 2172.89 | 2219.13 | 2189.17 | 2271.28 | 2291.80 | 2156.48 | 2210.29 |

but are also promising indicators that our method could scale to even larger datasets.

**Larger synthetic datasets.** To further investigate the efficiency gains of FIM over the baseline ConTinEst, for influence estimation, we also generated a synthetic dataset, Core$_{4096}$, with 4096 nodes and 10000 cascades. The corresponding MAE and running time of the two tested methods are in Table 7. (Recall that both NetRate and NMF are OOM in our computing environment for networks of this scale.) First, we observe the IE MAEs of FIM are consistently smaller than those of ConTinEst, with a difference ranging from 11.70% to 33.98%, which is in line with the results of Figure 6.

Moreover, FIM clearly outperforms ConTinEst in terms of running time. In particular, the speedup of FIM over the state-of-the-art ConTinEst is up to $100 - 120\times$, so two orders of magnitude faster. These observations, along with the results in Figure 7, further support the scalability of our approach.

## 6 RELATED WORK

**Network Inference.** Gomez-Rodriguez et al. [12] establish a generative probabilistic model to calculate the likelihood of cascade data. They devise the NETINF approach to infer the network connectivity, by a submodular maximization, aiming to maximize the cascades' likelihood. Similarly, Gomez-Rodriguez et al. [11] set a conditional likelihood of transmission with parameter $\alpha_{i,j}$ for each pair of nodes $i, j$ and derive the corresponding survival and hazard functions to express the likelihood of a cascade. They propose the algorithm NetRate to maximize the likelihood of cascades, by optimizing the pairwise parameter $\alpha_{i,j}$, which is the indicator of existence for edge $\langle i, j \rangle$. To capture heterogeneous influence among nodes (instead of following a fixed, parametric form), Du et al. [10] adopt a linear combination of multiple parameterized kernel methods to approximate the hazard function for the cascade likelihood maximization. This approach is shown to be more expressive than previous models. Later, Gomez-Rodriguez et al. [13] develop a more general additive and multiplicative risk model by adopting survival theory. As a result, the network inference problem is solved via convex optimization. We did not include the methods of Du et al.

(a) Time window $T$

(b) Time interval $\varepsilon$

**Figure 8: Running time of FIM on Weibo and Twitter.**

[10], Gomez-Rodriguez et al. [13] in our experimental comparison, as we were not able to obtain their implementation.

**Influence Estimation.** Du et al. [9] explores the influence estimation problem when the underlying network and transmission parameters are accessible. They point out that the set of influenced nodes is tractable through the shortest-path property. Based on this finding, they devise a novel size estimation method, by using a randomized sampling method from [6] and develop the ConTinEst algorithm for influence estimation. Later, Du et al. [8] further study the estimation of transmission parameters when only the network and cascade data are available. They propose InfluLearner to learn the diffusion function – using a convex combination of random basis functions – directly from cascade data, by maximizing the likelihood. However, InfluLearner requires knowledge of the cause (source node) of each activation in the cascades. Instead of focusing on the global influence of a seed set, Qiu et al. [33] aims to predict the local social influence of each individual user. To this end, they design an end-to-end framework called DEEPINF to learn latent representations, by incorporating both the network topology and user-specific features. Recently, He et al. [18] adopt neural mean-field dynamics to design NMF to approximate continuous-time diffusion processes. As shown by our experiments, our model FIM outperforms NMF in terms of both efficiency and accuracy, for the problems of network inference and influence estimation.

## 7 CONCLUSION

In this paper, we revisit the problems of network inference and influence estimation from continuous-time diffusion cascades. We propose the framework FIM, with the objective of improving upon both the learning ability and the scalability of existing methods. To this end, we first propose a continuous-time dynamical system to model diffusion processes, and build our framework FIM for network inference based on it. Furthermore, we improve upon the influence estimation by proposing a new sampling technique. We analyze the approximation error of FIM for network inference and the effect thereof on influence estimation. Comprehensive experimental results demonstrate the state-of-the-art performance of FIM for both network inference and influence estimation, as well as its superior scalability and applicability to real-world datasets. As a future work, we intend to extend and adapt FIM to different diffusion functions, besides the exponential one.

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

# A APPENDIX

## A.1 Proofs

PROOF OF THE LEMMA 1. For an observation $\mathbf{h}_t$, let $\mathcal{N}_v^a = \{u \mid \phi_{t,u} = 1 \ \forall u \in \mathcal{N}_v\}$ be the set of active neighbors of node $v$ at time $t$. Let $t_i$ be the time when the neighbor $u_i \in \mathcal{N}_v^a$ succeeds in activating node $v$ individually conditioned on $\mathbf{h}_t$ for $i \in \{1, 2, \ldots, |\mathcal{N}_v^a|\}$. Given a time interval $\tau$,

$$\Pr[\Phi_{t+\tau,v} = 1 \mid \mathbf{h}_t]$$
$$= 1 - \Pr\Big[\bigcup_{u_i \in \mathcal{N}_v^a} \{t_i > t + \tau \mid t_i > t\}\Big]$$
$$= 1 - \Pr\Big[\bigcup_{u_i \in \mathcal{N}_v^a} \{t_i > t + \tau, t_i > t \mid t_i > t\}\Big]$$
$$= 1 - \prod_{u_i \in \mathcal{N}_v^a} \frac{e^{-\lambda_{u_i v}(t+\tau)}}{e^{-\lambda_{u_i v} t}}$$
$$= 1 - \prod_{u_i \in \mathcal{N}_v^a} e^{-\lambda_{u_i v} \tau}$$
$$= 1 - e^{-\tau \sum_{u_i \in \mathcal{N}_v^a} \lambda_{u_i v}},$$

When $\tau \to 0^+$ (approaches 0 from the positive side) we have $\gamma_{t,v} = \sum_{u \in \mathcal{N}_v} \lambda_{uv} \phi_{t,u}$. □

PROOF OF LEMMA 2. Notice that the probability for a random node $v \in \mathcal{V}$ being inactive at time $t$ is $1 - \mathbb{E}[\Phi_{t,v} \mid \mathcal{H}_t]$. According to Lemma 1, we have

$$\Gamma_{t,v} = \mathbb{E}\Big[(1 - \Phi_{t,v}) \sum_{u \in \mathcal{N}_v} \lambda_{uv} \cdot \Phi_{t,u} \mid \mathcal{H}_t\Big].$$

For node set $\mathcal{V}$, it is straightforward to obtain Equation (2), which completes the proof. □

PROOF OF THEOREM 1. Equation (4) is proved in Lemma 2 and Equation (5) is the initial state given set $\mathcal{S}$. In what follows, we then prove Equation (3). Before that, we first establish the following ordinary differential equation.

$$\frac{d\mathbb{E}[\Phi_t \mid \mathcal{H}_t]}{dt} = \Gamma_t \tag{11}$$

First, for any $v \in V$, we have

$$\frac{d\mathbb{E}[\Phi_{t,v} \mid \mathcal{H}_t]}{dt}$$
$$= \lim_{\tau \to 0^+} \frac{\mathbb{E}[\Phi_{t+\tau} - \Phi_t \mid \mathcal{H}_t]}{\tau}$$
$$= \lim_{\tau \to 0^+} \frac{\Pr[\mathbf{t}_v \in [t, t+\tau] \mid \mathbf{t}_v \notin (0, t)]}{\tau}$$
$$= \lim_{\tau \to 0^+} \frac{\Pr[\mathbf{t}_v \in [t, t+\tau], \mathbf{t}_v \notin (0, t)]}{\Pr[\mathbf{t}_v \notin (0, t)] \tau}$$
$$= \lim_{\tau \to 0^+} \frac{\Pr[\mathbf{t}_v \in [t, t+\tau]]}{\Pr[\mathbf{t}_v \ge t] \tau}$$
$$= \lim_{\tau \to 0^+} \frac{\Gamma_{t,v} e^{-\Gamma_{t,v} t} \cdot \tau}{e^{-\Gamma_{t,v} t} \cdot \tau} \tag{12}$$
$$= \Gamma_{t,v}$$

According to the Euler method, we could derive Equation (3) from Equation (11). □

PROOF OF THEOREM 2. Given observation $\mathcal{H}_{t_0}$ at time $t_0$ and $v \in \mathcal{V}$, we have

$$\mathbb{E}[\Phi_{t_0+\varepsilon, v} \mid \mathcal{H}_{t_0}]$$
$$= \mathbb{E}[\Phi_{t_0, v} \mid \mathcal{H}_{t_0}] + \sum_{i=1}^{\infty} \frac{\mathbb{E}[\Phi_{t_0, v}^{(i)} \mid \mathcal{H}_{t_0}]}{i!} \varepsilon^i$$

where $\Phi_{t,v}^{(i)}$ is the $i$-th derivative of $\Phi_{t,v}$. Since we have $\mathbb{E}[\Phi_{t,v}^{(1)} \mid \mathcal{H}_{t_0}] = \Gamma_{t_0,v} e^{-\Gamma_{t_0,v}(t-t_0)}$. In this regard, we have

$$\mathbb{E}[\Phi_{t,v}^{(i)} \mid \mathcal{H}_{t_0}] = (-1)^{i+1} \Gamma_{t_0,v}^i e^{-\Gamma_{t_0,v}(t-t_0)}.$$

Therefore, we have

$$\xi_v(t_0, \varepsilon) = \varepsilon \Gamma_{t_0,v} - \sum_{i=1}^{\infty} \frac{\mathbb{E}[\Phi_{t_0,v}^{(i)} \mid \mathcal{H}_{t_0}]}{i!} \varepsilon^i$$
$$= \varepsilon \Gamma_{t_0,v} + \sum_{i=1}^{\infty} \frac{(-\varepsilon \Gamma_{t_0,v})^i}{i!}$$
$$= \varepsilon \Gamma_{t_0,v} + \sum_{i=0}^{\infty} \frac{(-\varepsilon \Gamma_{t_0,v})^i}{i!} - 1$$
$$= \varepsilon \Gamma_{t_0,v} + e^{-\varepsilon \Gamma_{t_0,v}} - 1$$

□

LEMMA 4 (HOEFFDING INEQUALITY [20]). *Let $x_i$ be an independent bounded random variable such that for each $1 \le i \le \theta$, $x_i \in [a_i, b_i]$. Let $X = \frac{1}{\theta} \sum_{i=1}^{\theta} x_i$. Given $\eta > 0$, then*

$$\Pr[|X - \mathbb{E}[X]| \ge \eta] \le 2e^{-\frac{2\theta^2 \eta^2}{\sum_{i=1}^{\theta} x_i (b_i - a_i)^2}}.$$

PROOF OF THEOREM 3. Given seed set $\mathcal{S}$ and diffusion rate vector $\rho^*$, let $O$ be the sampling space of instance $\mathcal{G}$ of G. For instance $\mathcal{G} \in O$, let $\ell$ be the shortest path in $\mathcal{G}$ from certain source node $u_0 \in \mathcal{S}$ to node $u_k \in \mathcal{V}$ within time $T^*$, i.e., $\ell = \{\langle u_i, u_{i+1} \rangle \mid i \in \{0, 1, \ldots, k-1\}\}$. Let $\lambda_i^* \in \rho^*$ be the diffusion rate and $t_i^*$ be the sampled diffusion time for edge $\langle u_i, u_{i+1} \rangle$. For estimation $\rho$, the corresponding diffusion rate and diffusion time are $\lambda_i$ and $t_i$ respectively. We consider two cases.

**Case one:** $\lambda_i \in [\lambda_i^*, \lambda_i^* + \epsilon]$. In this case, $t_i$ can be lower bounded by samplings from $\min(t_i^*, p(\epsilon, t))$. Thus we have

$$\mathbb{E}[t_i] \ge \int_0^{t_i^*} t p(\epsilon, t) dt + \Big(1 - \int_0^{t_i^*} p(\epsilon, t) dt\Big) t_i^*$$
$$= \int_0^{t_i^*} t \cdot \epsilon e^{-\epsilon t} dt + \Big(1 - \int_0^{t_i^*} \epsilon e^{-\epsilon t} dt\Big) t_i^*$$
$$= -t_i^* e^{-\epsilon t_i^*} - \frac{e^{-\epsilon t_i^*} - 1}{\epsilon} + t_i^* e^{-\epsilon t_i^*} \epsilon$$
$$= \frac{1 - e^{-\epsilon t_i^*}}{\epsilon}$$

Since $\frac{\mathbb{E}[t_i]}{t_i^*} \ge \frac{1 - e^{-\epsilon t_i^*}}{\epsilon t_i^*} \ge \frac{1 - e^{-\epsilon T^*}}{\epsilon T^*}$, we have $\frac{\mathbb{E}[T]}{T^*} \ge \frac{1 - e^{-\epsilon T^*}}{\epsilon T^*}$.

**Case two:** $\lambda_i \in [\lambda_i^* - \epsilon, \lambda_i^*]$. The probability $p_i$ that $u_i$ reaches $u_{i+1}$ in instance $\mathcal{G}$ is $p_i = \int_0^{t_i^*} p(\lambda_i^*, t) dt$. Therefore, for estimation $\lambda_i$ and $t_i$, it requires $\int_0^{t_i^*} p(\lambda_i^*, t) dt = \int_0^{\mathbb{E}[t_i]} p(\lambda_i, t) dt$. It is easy to obtain that $\frac{\mathbb{E}[t_i]}{t_i^*} = \frac{\lambda_i^*}{\lambda_i} \le \frac{\lambda_i^*}{\lambda_i^* - \epsilon} \le \frac{\min(\rho^*)}{\min(\rho^*) - \epsilon} = \frac{c}{c-1}$, which completes the proof. □

## A.2 Other Experimental Details

*A.2.1 Publicly available datasets.* **MemeTracker.** Meme-Tracker [26] is a dataset aiming to track the diffusion of memes, e.g., frequent quotes or phrases, in online websites and blogs over time. Specifically, it collects a million news stories and blog posts and then selects the most frequent quotes and phrases as "memes". By regarding each meme as an information item and each URL or blogs as a user, it tracks the diffusion of memes.

**Sina Weibo.** Sina Weibo [41] is a major micro-blogging platform in China. With an underlying follower - followee network, Weibo users can share news or comments with timestamps and their followers can repost the content. By tracking the diffusion of each post, we have continuous-time cascades.

**Twitter.** Similar to Weibo, Twitter [19] is the most popular micro-blogging application. With an underlying follower-followee network, each tweet with URLs is taken as a node and all tweets with URLs posted during October 2010 are collected in Twitter dataset which contains the complete tweeting trace of each post during that period.

**Pre-processing.** Following the methodology of previous works [11, 12], we only keep the nodes that are the most involved in diffusions, in order to refine the quality of the cascade data for network inference and influence estimation. Specifically, a node is seen as actively participating in diffusions if it appears in at least 5 cascades. We remove the cascades that do not contain such nodes.

**Cascades.** One cascade $\mathbf{c}$ consists of activation timestamps of all the participating nodes. Specifically, cascades for the above three real-world datasets were collected online. Cascades for the HR dataset are provided by [18] and the cascades of the 6 synthetic datasets were generated by simulating the diffusion process. One particularity is that the HR cascades may contain multiple source nodes, i.e., several nodes with active times equal to 0, while the cascades of all the other datasets can only have one source node.

*A.2.2 Experimental setting.* **Computing environment.** All experiments are conducted on a machine with an NVIDIA RTX A5000 GPU (24GB memory), AMD EPYC CPU (1.50 GHz), and 500 GB of RAM.

**Source code of FIM and baselines.** During the open review phase of the paper, we will make the source code of FIM publicly accessible through an anonymous GitHub link. For baseline methods, we downloaded their implementation from the official releases, i.e., NetRate[3], ConTinEst[4], and NMF[5].

**Ground-truth spread.** By following the convention [8, 18], the ground-truth spread of a source set $\mathcal{S}$ is evaluated as follows. Given a source set $\mathcal{S}$, we first randomly sample from the test cascades a cascade for each node in $\mathcal{S}$, and then take the union of the nodes from the sampled cascades, leading to one estimate of $\mathbb{E}[\mathcal{S}]$. We repeat this process 1000 times and compute the average over estimates as the ground-truth spread.

**Motivation and rationale behind IM experiments.** Besides the binary cross-entropy (BCE) loss, we designed an additional

---

[3]https://github.com/Networks-Learning/netrate
[4]https://dunan.github.io/DuSonZhaMan-NIPS-2013.html
[5]https://github.com/ShushanHe/neural-mf

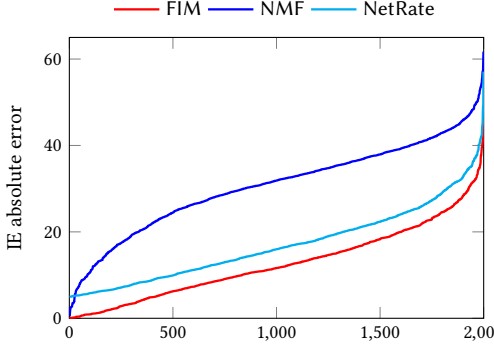

**Figure 9: IE absolute error on HR.**

experiment, for CIM, to further verify the quality of the estimated parameter matrix $\mathbf{A}$. The rationale is the following. For the sake of the example, let $\mathbf{A}_{\mathsf{FIM}}$ and $\mathbf{A}_{\mathsf{NMF}}$ be the estimated parameter matrices by FIM and NMF respectively, and let $\mathbf{A}^*$ be the true parameter matrix (unknown). W.l.o.g, let $\mathcal{S}_{\mathsf{FIM}}$ and $\mathcal{S}_{\mathsf{NMF}}$ be the seed sets of the same sizes selected by the state-of-the-art CIM algorithm on $\mathbf{A}_{\mathsf{FIM}}$ and $\mathbf{A}_{\mathsf{NMF}}$ respectively, and $\mathcal{S}^*$ be the unknown optimal seed set (i.e., with the largest expected spread) on $\mathbf{A}^*$ (were it known). For any selected seed set $\mathcal{S}$, its *ground-truth spread* $\mathbb{E}[I(\mathcal{S})]$ on $\mathbf{A}^*$ can be estimated using the testing cascades, as described previously. If we observe that $\mathbb{E}[I(\mathcal{S}_{\mathsf{NMF}})] < \mathbb{E}[I(\mathcal{S}_{\mathsf{FIM}})]$, this means $\mathbb{E}[I(\mathcal{S}_{\mathsf{FIM}})]$ is closer to $\mathbb{E}[I(\mathcal{S}^*)]$ since $\mathcal{S}^*$ has by definition the largest expected spread. This indicates that $\mathbf{A}_{\mathsf{FIM}}$ is closer to the optimal $\mathbf{A}^*$, hence represents further proof for the higher quality in the estimation of $\mathbf{A}$.

## A.3 Additional Experimental Results

**Evaluation of NetRate for parameter learning.** We assess in this experiment NetRate's performance on HR dataset, from the perspective of influence spread. Since HR's cascades have multiple source nodes, we cannot directly use the ground-truth spread estimation. As a workaround, we use the validation and test cascades (2000 cascades) as follows. We take the node set of each cascade $\mathbf{c}$ within the time window $T = 10$ as the ground-truth spread of the source set $\mathcal{S}$. Subsequently, we run FIM, NMF, and NetRate for influence estimation (IE) based on their own estimated $\mathbf{A}$ and then compare the IE absolute error of each cascade. Figure 9 plots the IE absolute error in non-decreasing order, by the three tested methods. As shown, FIM achieves notably smaller absolute error than the other two methods. In particular, the mean absolute error (MAE) for FIM, NMF, and NetRate are 12.42, 28.86, and 17.90, respectively. In other words, the MAE of FIM is significantly lower than the one of both NMF and NetRate, representing only 43.05% and 69.43% of their respective values. This result further confirms the performance of FIM on network inference.

**Additional IM experimental results on MemeTracker.** Recall that Table 4 in Section 5.1 presents the influence maximization results of FIM and NMF on the MemeTracker data, with a time window $T = 10$ and a time interval $\varepsilon = 1$. For a comprehensive evaluation, we present here additional results with two more $(T, \varepsilon)$ pairs, $(T = 10, \varepsilon = 0.5)$ and $(T = 15, \varepsilon = 1.0)$ in Table 8. Overall,

**Table 8: Additional influence maximization on MemeTracker.**

| Parameters $T$ and $\varepsilon$ | Method | Seed Set Size $|\mathcal{S}|$ | | | | | | |
|---|---|---|---|---|---|---|---|---|
| | | 4 | 5 | 6 | 7 | 8 | 9 | 10 |
| $T = 10, \varepsilon = 1.0$ | FIM | 65.85 | 75.18 | 88.33 | 108.88 | 113.90 | 127.11 | 136.67 |
| | NMF | 53.14 | 64.78 | 70.66 | 76.10 | 83.39 | 88.23 | 94.95 |
| $T = 10, \varepsilon = 0.5$ | FIM | 69.53 | 76.88 | 92.70 | 101.79 | 122.58 | 135.04 | 144.66 |
| | NMF | 47.24 | 53.40 | 68.88 | 75.09 | 68.99 | 87.20 | 80.74 |
| $T = 15, \varepsilon = 1.0$ | FIM | 67.59 | 72.85 | 88.88 | 96.37 | 103.35 | 107.71 | 119.41 |
| | NMF | 61.00 | 68.60 | 76.82 | 86.56 | 84.32 | 91.26 | 95.74 |

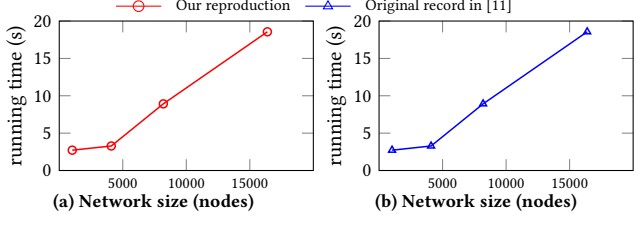

**Figure 10: The reproduction of running time of NetRate.**

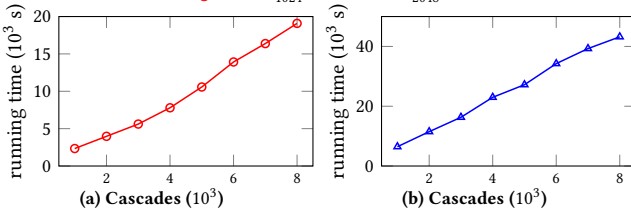

**Figure 11: The Running time on different cascades of NetRate.**

seed sets selected from the estimated **A** by FIM lead clearly to larger influence spread than those of NMF. Compared with the results for the combination ($T = 10, \varepsilon = 1.0$), the spread of FIM in the ($T = 10, \varepsilon = 0.5$) setting increases slightly, as expected, since smaller $\varepsilon$ means smaller estimation error, as proved in Theorem 2. Furthermore, we can observe that the spread of FIM in the setting ($T = 15, \varepsilon = 1.0$) decreases, especially when $|\mathcal{S}| = \{7, 8, 9, 10\}$. This peculiar outcome is likely due to the insufficient number of cascades, which leads to an insufficiently precise estimation of the ground-truth influence spread.

**NetRate reproduction of experimental results.** Recall that NetRate approaches network inference from continuous-time cascades as a convex optimization problem and utilizes CVX as the problem solver [17]. Specifically, NetRate creates an individual convex problem for each node. The number of variables of each convex problem is linear in the number of nodes and cascades, as well as in the size of each cascade. Therefore, as the number of nodes increases, more

new convex problems are generated. As the number of cascades increases, more variables are introduced in the objective functions. Therefore, both facets contribute to a very high computational time on reasonably small problem instances.

As a sanity check for our experimental setting, following the experimental framework described in [11], we reproduce here the experiments of NetRate (Figure 3(c) in [11]) and present the reproduced results in Figure 10. Specifically, Figure 10(a) shows our reproduction of NetRate 's average running time to infer transmission rates for all incoming edges to a node against network size (number of nodes), compared with the original results in Figure 10(b). The high similarity between these two sub-figures confirms the fidelity of our reproduction and testing setting for NetRate. Moreover, we further investigate NetRate's performance in Figures 11(a) and (b), on the datasets $Rand_{1024}$ and $Rand_{2048}$, depending on the number of cascades. The results reveal once more that NetRate's computational time increases drastically as the input size increases.

