# OpenReview forum: "Scalable Continuous-time Diffusion Framework for Network Inference and Influence Estimation"
_ACM.org/TheWebConf/2024/Conference — TheWebConf24 Oral_

### Official Review · Reviewer_SU74 · 2023-11-24

**Novelty:** 5
**Technical Quality:** 4

**Review:**

The paper presents a novel framework named FIM (Framework of dIffusion approxiMation) designed to tackle the computational complexities associated with large-scale network inference and influence estimation. This framework is grounded in the view of the diffusion process as a continuous-time dynamical system, which is a significant shift from traditional discrete models. The author first formalized the diffusion graph problem under continuous time conditions, then introduced the model framework, and conducted experiments on real world datasets and self generated datasets.

Strengths:

1. Previous methods for solving diffusion graph problems have treated the diffusion process of the graph as discrete, which may not be entirely consistent with the situation in the real world. The author's transformation of the diffusion graph problem into a continuous time range for consideration is a very important innovation.

2. The model proposed by the author can utilize small computational power for large-scale data calculations, greatly improving the scope of use of graph diffusion algorithms for big data.

Weaknesses:

1. The baseline used in the article is too outdated, and the optimization results generated compared to the baseline cannot prove its superiority over the latest methods.

2. The approximation errors inherent in the model are analyzed, and an advanced sampling technique, shortest diffusion time of set (SDTS), is introduced to enhance efficiency in influence estimation. However, the atricle lacks necessary ablation experiments to demonstrate the effectiveness of SDTS.

3. The method in the article is based on the assumption that the current network architecture will not change. This limits the effectiveness of extending this method to actual datasets. In fact, network architectures such as social networks are constantly changing, especially when the time span is extended.

4. The real-world datasets used in the experiment, such as Twitter, Weibo, etc., are all social media datasets. The author should explore the scalability of the model using datasets from different fields.

**Questions:**

1. How much impact does SDTS have on the model?

2. Table 6 shows the BCE loss of FIMs on two real-world datasets. The author demonstrated the availability of FIMs on large networks in section 5.3, but Table 6 is not an effective proof method as there are no other models compared to FIMs. In addition, there is a significant difference in BCE loss between the two datasets. Does this indicate different features in the Weibo and Twitter datasets?

3. Can the FIM framework be adapted for different types of networks, such as weighted or directed networks?

4. The diffusion process is calculated based on the initial state S, will the initial state S have a significant impact on the final result? If the impact is significant, will it lead to bias in the results for larger time spans, especially for future diffusion processes (when the initial state needs to be determined based on the FIM model)?

**Reviewer Confidence:**

3: The reviewer is confident but not certain that the evaluation is correct

**Scope:**

4: The work is relevant to the Web and to the track, and is of broad interest to the community

---

### Official Review · Reviewer_J8EE · 2023-11-25

**Novelty:** 6
**Technical Quality:** 6

**Review:**

The paper studies the information diffusions with the two fold goal. First, it develops an effective framework for inferring the underlying network structure from the information about cascade evolution. Second, is uses the estimated network structure for influence estimation. The paper proposes a rather natural framework in this context network is inferred using SGD with respect to the binary cross-entropy computed over the cascade evolution for a given network structure. In fluence estimation is on the other hand done using the concept of shortest diffusion time.  Finally, the paper compares the newly proposed algorithm with two baseline approaches NMF and NetRate and shows that it indeed leads to lower error in the estimation. The paper is well written and explains the introduced concepts well. I think that it is worth to be accepted as it advances the state-of-art.

**Questions:**

I do not have any specific question.

**Ethics Review Description:**

not required

**Reviewer Confidence:**

3: The reviewer is confident but not certain that the evaluation is correct

**Scope:**

4: The work is relevant to the Web and to the track, and is of broad interest to the community

---

### Official Review · Reviewer_dAzg · 2023-11-26

**Novelty:** 5
**Technical Quality:** 5

**Review:**

This paper studies the problem of network inference and influence estimation only from continuous-time cascades. The authors propose the FIM framework, both improve the learning ability and the scalability of existing methods. They verify the effectiveness and efficiency of FIM by synthetic and real-world datasets.

S1. The study of continuous-time diffusion has a wide range of web applications, and this paper extends the problem to larger real-world datasets with full practical implications.
S2. This paper proposes a novel sampling technique, which greatly improves the efficiency of the diffusion influence estimation. The theoretical part of the methods is solid and clear.
S3. The authors conducted thorough experiments, the results of which further ensured the accuracy and computational efficiency of the framework. The authors also experimented with a rich set of evaluation metrics, illustrating the excellence of the algorithm's parameter learning ability.
S4. The paper is well organized, and it is very easy to follow the concepts and symbols.

W1. In the Introduction, the authors propose to view the process of influence diffusion as a continuous-time dynamical system, and this section lacks additional explanation of continuous-time dynamical system and concept particle.
W2. In section 3.3, the time complexity of Equation 4 is considered to be O(m), in fact, the complexity of Equation 4 is related to the number of non-zero elements in the adjacency matrix A. The complexity of this operation is O(m) only if the original network is a sparse graph. Therefore, should that part possibly be modified to O(n^2)?
W3. Baseline methods are not constructed using BCE loss as their evaluation metric, so it may be unfair to compare the performance of FIM and baseline using BCE loss.

**Questions:**

Q1: Can the authors provide more details about the SDTS method? For example, how to sample the transmission time $t_{u, v}$ from the edge $<u, v>$.
 Q2: From the experimental results in Table 6, it can be seen that the BCE loss results on the Twitter dataset are significantly smaller than the results on the other datasets, is this bias in the results caused by the fact that this dataset has the highest number of nodes but lowest number of cascades?
 Q3: When inferring the structure of the network, the authors use the SGD+BP approach to update the parameters of the adjacency matrix A. Can the authors provide relevant proof that the final result can converge to the optimal value?

**Reviewer Confidence:**

3: The reviewer is confident but not certain that the evaluation is correct

**Scope:**

3: The work is somewhat relevant to the Web and to the track, and is of narrow interest to a sub-community

---

### Official Review · Reviewer_s1Zd · 2023-11-27

**Novelty:** 5
**Technical Quality:** 5

**Review:**

The papers addresses several problems in influence estimation and network inference. The goal is to improve upon the applicability and scalability of existing methods. To this end, they propose a framework (FIM) which estimates edge diffusion parameters based on available cascade information, by modeling the process as a set of differential equations which is then discrete-time approximated. Besides network inference, they also show how the FIM framework can be applied to influence estimation, by both theoretically bounding approximation guarantees and empirical verification.

The paper is very solid, well-written and all the claims are backed up with sufficient theoretical proofs and intuitive reasoning.
I think this paper makes a nice addition to the field, and especially considering the improvement in scalability while achieving good performance.

minor:
- line 213: the definition of cascade c is incorrect. It should be c = {t_u: u \in V}. Not 'for all'.
- line 217, same as the comment above.

**Questions:**

The choice of time interval 𝜀 seems to play an important role.
How does 𝜀 influence running time? Is there any experiment that shows this in more detail besides Fig.3 and Fig.4?
Did you observe an significant increase in performance when setting 𝜀 below 0.5?

**Ethics Review Description:**

no ethical issues

**Reviewer Confidence:**

2: The reviewer is willing to defend the evaluation, but it is likely that the reviewer did not understand parts of the paper

**Scope:**

4: The work is relevant to the Web and to the track, and is of broad interest to the community

---

### Decision · Program_Chairs · 2024-01-22

**Decision:**

Accept (Oral)

**Comment:**

This paper seeks to improve algorithms for inferring both network structure and influence in (continuous-time) settings of information cascades, through a continuous-time dynamical system lens. The resulting algorithms (the "FIM" framework) are analyzed both theoretically and empirically, and generally outperform existing approaches.

 The reviewers consistently praise the the technical contributions of this submission, and its improvements on the existing algorithms in the literature. They also speak highly of the writing/presentation of this work, including the overall organization of the paper, as well as both the proofs and the more intuitive explanations within it. The concerns raised by the reviewers tended to be smaller: some expository points (e.g., some underdescribed elements), questions about whether the baselines to which comparisons were made are fair, and some smaller technical concerns. The authors' response to technical questions from the reviewers was generally perceived as helpful, and they suggested edits to the paper in response to the questions asked by the reviewers that seem to address some of the more significant concerns. (Not all reviewers were able to respond to the authors' replies.) The consensus of the reviewers was positive in most ways.